

# Psychological distress and problematic internet use among language teachers: a latent profile analysis

Zizheng Shen[1], Honggang Liu[1] and I-Hua Chen[2,3,4]

[1] School of Foreign Languages, Soochow University, Suzhou, Jiangsu, China
[2] Chinese Academy of Education Big Data, Faculty of Education, Qufu Normal University, Qufu, Shandong, China
[3] Jiangxi Psychological Consultant Association, Nanchang, Jiangxi, China
[4] Graduate School, Stamford International University of Thailand, Bangkok, Thailand

## ABSTRACT

**Background.** Given the limited attention to psychological distress (PD) and problematic internet use (PIU) among language teachers, the study tried to reveal the different PD profiles and their association with PIU.

**Methods.** We first employed latent profile analysis (LPA) to identify the latent profiles within a cohort of language teachers in China and then utilized multivariate logistic regression analysis to explore the demographic characteristics associated with different PD profiles. After that, one-way analysis of variance (ANOVA) was conducted to examine the relationships between PD and PIU.

**Results.** The LPA identified three distinct latent profiles of PD: the moderate-to-severe PD profile, the mild PD profile, and the mentally healthy profile. The results of multivariate logistic regression indicated that male and experienced language teachers were more likely to experience severe PD. The results of one way ANOVA suggested that language teachers experiencing PD reported notably higher levels of PIU, with those in the moderate-to-severe PD group scoring particularly high in internet gaming disorder.

**Discussion.** Results of this study indicate the close association between PD and PIU in language teachers. These findings underscore the critical need to address PD among language teachers and emphasize the importance of education and training aimed at promoting teachers' appropriate internet use. Breaking the "PD–PIU" vicious cycle is essential for fostering better mental health and well-being within this professional group.

# INTRODUCTION

Psychological distress (PD) has emerged as a critical global public health issue, particularly within occupational settings (*Chen, Qu & Hong, 2022*). The teaching profession demonstrates exceptional vulnerability to PD, with meta-analytic evidence documenting consistently elevated distress levels among teachers across diverse cultural contexts

Corresponding authors
Honggang Liu,
liuhonggang@suda.edu.cn
I-Hua Chen,
chenih0807@qfnu.edu.cn

(*Agyapong et al., 2022*; *Ozamiz-Etxebarria et al., 2021a*; *Ozamiz-Etxebarria et al., 2021b*). This professional susceptibility originates from the unique occupational ecology of teaching, which requires constant navigation of competing stakeholder expectations while maintaining work-life equilibrium (*Fu et al., 2023*). Such chronic role strain has been implicated in the progressive deterioration of teacher mental health across international educational systems (*Lizana & Lera, 2022*; *Zhou et al., 2024*). The Chinese context presents a special case for investigating educator PD, owing to its unique sociocultural and institutional characteristics. Recent nationwide data reveals that Chinese teachers experience mental health disorders at a rate of 16.1% (*Jin & Yu, 2024*), demonstrating more severe mental health impairment than other professional groups (*Fu et al., 2023*; *Johnson et al., 2005*). This pronounced mental health disparity underscores the imperative for investigations of teachers' PD in China's educational ecosystem.

Among teaching professionals, language educators may face compounded risks due to discipline-specific stressors. The dynamic nature of language instruction demands continuous cognitive engagement to facilitate spontaneous classroom interactions (*İpek et al., 2018*). In China's language education system, these challenges are exacerbated by the inherent conflict between communicative teaching approaches and examination-oriented curricular requirements (*Li, 2024*). Emerging evidence suggests that language teachers experiencing PD may increasingly rely on digital platforms for emotional regulation, potentially leading to problematic internet use (PIU) (*He & Chen, 2024*; *Lee & Chen, 2021*). This pattern points to a potentially significant intersection between PD and PIU that warrants systematic investigation.

However, extant research has predominantly used variable-centered approaches to investigate PD and its association with PIU, thereby overlooking meaningful sub-populations with unique PD profiles (*i.e.,* distinct teacher groups with different distress patterns and varying PIU relationships). To bridge these critical research gaps, the present study adopted latent profile analysis (LPA), a person-centered analytical framework, to achieve two key objectives: (1) identify distinct PD profiles among Chinese language teachers and their demographic characteristics; and (2) investigate how these profiles differently relate to PIU severity. This person-centered analytical approach offers distinct advantages over traditional methods by capturing the nuanced manifestations of distress within Chinese language teachers. The findings will contribute to theoretical understanding of occupation-specific mental health patterns while informing targeted intervention strategies for language educators in high-demand environments.

## LITERATURE REVIEW

### Teachers' psychological distress

Psychological distress is a multifactorial construct that refers to non-specific symptoms of depression, anxiety, or stress more generally (*Belay, Guangul & Asmare, 2020*; *Chen, Qu & Hong, 2022*; *Monteagudo et al., 2023*). High levels of distress can serve as a marker of impaired mental health (*Zhu et al., 2022*). Teaching is one of the most stressful professions (*Gadermann et al., 2023*; *Lizana & Lera, 2022*). *Johnson et al. (2005)* reported that teachers'

psychological health, physical health, and job satisfaction were lower than other professions like researchers, police, and medical workers. The increasing prevalence of mental health issues among teachers over recent decades, along with their broad social implications, has garnered global attention, highlighting the critical importance of preventing PD and fostering mental well-being in the teaching profession (*Agyapong et al., 2022*; *Liu, Li & Fang, 2024*; *Rojas-Andrade, Aranguren Zurita & Prosser Bravo, 2024*; *Yang et al., 2019*).

Although all teachers face growing pressures and complexity in their professional lives (*Day & Qing, 2009*), language teachers may suffer greater PD than their counterparts in other subjects (*Piechurska-Kuciel, 2011*). A survey conducted in China revealed that the prevalence of depression among foreign language teachers (17.1%) significantly exceeded that of math teachers (14.4%), while anxiety rates reached 54.7%—higher than those of math teachers (48.8%), and teachers of other subjects (49.1%) (*Fu et al., 2023*). The phenomenon may be particularly pronounced in primary and secondary education (*Ozamiz-Etxebarria et al., 2021a*; *Ozamiz-Etxebarria et al., 2021b*), where teaching autonomy and curricular flexibility are more restricted compared to higher education faculty (*Piechurska-Kuciel, 2011*; *Teven, 2007*).

The rising PD among language teachers may originate from unique challenges inherent in language teaching, such as linguistic anxiety (*Cardoso-Pulido, Guijarro-Ojeda & Pérez-Valverde, 2022*). In addition to these challenges, the technology-driven era has brought additional PD risks related to technology. First, the digital revolution has fundamentally altered their professional landscape. The proliferation of online learning resources has eroded teachers' traditional knowledge authority (*Cardoso-Pulido, Guijarro-Ojeda & Pérez-Valverde, 2022*; *Mercer, Oberdorfer & Saleem, 2016*), with many no longer perceiving themselves as the "sole source of knowledge and information" in their language classroom *Mercer, Oberdorfer & Saleem* (*2016*, p. 215). This paradigm shift constitutes what *Cardoso-Pulido, Guijarro-Ojeda & Pérez-Valverde (2022*, p.3) term a "powerful disruptive element" to language teacher well-being. Second, technological advancements, especially in artificial intelligence, present existential challenges to the profession (*Gao et al., 2023*; *Rashid & Kausik, 2024*), intensifying job insecurity. Compounding these pressures, global education reforms continue raising performance expectations (*Jiang, 2017*; *Li & Liang, 2024*), despite persistently low salaries (*Fu et al., 2023*; *Mercer, Oberdorfer & Saleem, 2016*). These cumulative stressors create a perfect storm for PD development among language educators. Language education is a crucial element of basic education (*Qi, 2016*; *United Nations Educational, Scientific and Cultural Organization, 2019*). The psychological well-being of language teachers can significantly impact their teaching practices as well as students' language learning experiences and outcomes (*Liu et al., 2025*; *Yang, 2022*; *Zhou et al., 2024*). Therefore, the PD faced by language teachers warrants comprehensive and in-depth investigation.

However, the majority of quantitative studies that investigated the teachers' PD, or teacher psychology more broadly, have incorporated a variable-centered approach, such as regression analysis, and structural equation modeling, among others (*e.g.*, *Cao et al., 2023*; *Lizana & Lera, 2022*; *Miconi et al., 2024*), treating teachers as a homogenous group. While these studies have yielded valuable insights into the factor structures, antecedents,

and consequences of teachers' PD, they have largely overlooked the inherent heterogeneity within this professional group. The population of teachers is heterogeneous (*Lassri, 2023*), so individual differences may play an important role in the varied responses to PD items.

Previous research (*e.g.*, *Li et al., 2020*; *Lizana & Lera, 2022*; *Ozoemena et al., 2021*) has documented that several demographic characteristics could predict the degree of teachers' PD, such as gender, teaching experience, school stage, and so on. Female, novice, and primary school teachers have been recognized as being vulnerable to PD by most previous studies (*e.g.*, *Fernández-Berrocal et al., 2017*; *Ozoemena et al., 2021*). However, some recent studies reported different observations. For example, *Li et al. (2020)* reported that male teachers with ages 60 to 100 years and female with ages 50 to 60 had the highest prevalence of anxiety. *Ozoemena et al. (2021)* found that old experienced teachers have shown more psychological symptomatology than younger teachers, which might be due to the digital gap (*Ozamiz-Etxebarria et al., 2021a*; *Ozamiz-Etxebarria et al., 2021b*; *Song & Chen, 2019*).

Those inconsistent findings may be caused by neglecting the heterogeneity of teaching staff, warranting the necessity of using the personal-centered approach to clarify the predictive roles of these demographic variables on language teachers' PD. LPA is such a technique, that focuses on finding out the unobserved heterogeneity within a population and ultimately provides us with latent sub-populations within the population (*Asendorpf, 2015*). This technique provides two key methodological advantages for examining PD among language educators: first, it reveals nuanced symptom patterns that traditional variable-centered approaches often fail to detect (*Spurk et al., 2020*), enabling a more sophisticated understanding of distress manifestations. Second, it facilitates the identification of distinct high-risk subgroups within the teaching population (*Howard & Hoffman, 2018*), which is crucial for developing precisely targeted intervention strategies. These capabilities make LPA particularly valuable for investigating the complex psychological experiences of language teachers, who face unique occupational stressors that may produce heterogeneous distress profiles. While *Lassri (2023)* has utilized this technique to identify the latent profiles of PD among general teachers, this proposition has yet to be investigated among language teachers to the best of our knowledge. Therefore, this study employed LPA to identify the unobserved latent subgroups in a sample of Chinese primary and secondary school language teachers and further explore the demographic characteristics of different latent groups. We aspired to provide a more detailed understanding of the PD experienced by language teachers and offer some practical insights for educators and stakeholders, supporting the professional well-being of language teachers and ultimately benefiting practical language teaching.

## Teachers' psychological distress and problematic internet use

Problematic internet use, also internet addiction (IA), refers to an inability to control one's use of the internet which leads to negative consequences in daily life (*Spada, 2014*). In this study, the construct is conceptualized as the extended, compulsive utilization of the internet (*He & Chen, 2024*), rather than being classified as a pathological condition. It characterizes an uncontrollable urge to engage with the internet, perceiving periods, devoid of the internet access as insignificant, and manifesting excessive irritability and

aggression when access is impeded (*Young, 2004*). PIU specifically encompasses issues such as excessive social media use, smartphone addiction, and problematic gaming behavior. Prior studies have highlighted the detrimental effects of PIU on individuals' lives and work, including poor sleep quality, reduced physical activity, neglect of family responsibilities, job performance decline, and withdrawal from social interactions (*He & Chen, 2024*; *Kapus et al., 2021*; *Xu et al., 2021*). These consequences are particularly concerning within the teaching profession, where PIU may disrupt pedagogical processes and negatively impact students' learning experiences and academic achievement (*He & Chen, 2024*).

While existing research has extensively documented PIU among adolescents (*Cao & Su, 2007*; *Lozano-Blasco, Robres & Sánchez, 2022*), far less attention has been paid to PIU among teachers, despite their critical role in shaping students' digital behaviors (*Szymkowiak et al., 2021*). This gap is particularly striking in the case of language teachers, whose professional context may heighten their vulnerability to PIU (*He & Chen, 2024*). Two key factors underscore the urgency of investigating PIU in this sub-population. First, as one of primary role models, language teachers' digital habits can influence students' technology use patterns (*Lai, Li & Wang, 2017*; *Szymkowiak et al., 2021*). Second, the profession itself entails unique risk factors: (1) the highly interactive nature of language teaching, coupled with its emotional and cross-cultural demands, often leads to psychological fatigue (*Horwitz, 1996*; *İpek et al., 2018*; *Mercer, Oberdorfer & Saleem, 2016*; *Wang & Song, 2022*), potentially driving compensatory overuse of online platforms; and (2) in today's technology-integrated educational landscape, the increasing reliance on digital tools (*Mercer, Oberdorfer & Saleem, 2016*), such as vocabulary apps, video-based materials, and online professional development resources, obscures the line between essential and excessive use, while simultaneously complicating the differentiation between productive use and problematic overuse. Given their dual role as behavioral exemplars and a high-risk occupational group, language teachers represent a critical yet understudied population in PIU research. Addressing this gap is essential not only for safeguarding educators' well-being but also for ensuring their effectiveness in fostering healthy digital habits among students.

Previous research has demonstrated a robust association between PD and PIU across diverse populations. Among adolescent and adult cohorts, individuals with internet addiction consistently exhibit elevated levels of depression, anxiety, and stress compared to their non-addicted counterparts (*Ostovar et al., 2016*). This correlation is particularly pronounced in smartphone overuse, where depressive and anxious symptomatology serve as significant predictive factors (*Rho et al., 2019*). Teacher population demonstrates similar vulnerability to this association. Studies from different contexts highlighted that teachers experiencing PD face an increased risk of developing PIU (*Lee & Chen, 2021*; *Tsumura et al., 2018*; *Yi et al., 2021*). For instance, research conducted in Japanese educational settings revealed that schoolteachers with PD reported significantly higher rates of PIU, with consistent patterns observed across genders (*Tsumura et al., 2018*). Parallel findings have emerged in Chinese contexts: studies by *Lee & Chen (2021)* and *Yi et al. (2021)* further corroborated strong correlations between teachers' PD levels and PIU tendencies.

Evidence suggests a bidirectional relationship between these two constructs (*Odacı& Çıkrıkçı, 2022*). On the one hand, PD is a major contributor to PIU. The internet for those frustrated teachers may be a sanctuary from stress and cruelty in reality (*Lee & Chen, 2021*; *Tsumura et al., 2018*). Thus, teachers suffering from PD are more likely to engage in PIU (*Kızılok & Özok, 2021*; *Young, 2017*). On the other hand, overuse of the Internet will exacerbate emotional perception (*Gao & Chen, 2006*; *Liao et al., 2023*), making people more sensitive, and leading to sleep problems (*He & Chen, 2024*). Poor sleep, further, amplifies basic emotional reactivity, increasing negative mood states (*e.g.*, anxiety, depression, suicidality) (*Ben Simon et al., 2020*), and intensifying the deterioration of individuals' mental health. The educational implications of this vicious cycle are particularly concerning. As key factors in students' proximal environment (*Pekrun, 2006*), language teachers' mental health and behavioral patterns significantly influence adolescent development (*Cao et al., 2023*; *Li, Duan & Liu, 2023*). Their compromised psychological well-being may create ripple effects throughout the language classroom environment, ultimately affecting students' socioemotional growth and language academic achievement (*Liu & Li, 2023*).

Despite these insights, the interplay between PD and PIU remains underexplored among language teachers, particularly in primary and secondary education. To address this gap, the present study employed one-way analysis of variance (ANOVA) to examine differences in the degree of PIU across latent groups with varying levels of PD. By providing empirical evidence, this study aims to deepen the understanding of the complex interplay between teachers' PIU and PD. It also seeks to offer practical guidance for preventing and addressing PD and PIU among language teachers. In summary, our research questions are as follows:

(1)  What are the PD profiles of language teachers in primary and secondary schools?
(2)  What are the demographic characteristics of language teachers with different profiles?
(3)  What are the relationships between PD profiles and teachers' PIU?

## RESEARCH DESIGN

### Participants

A non-probability sampling (convenience sampling) was used to recruit our participants. We recruited our participants from more than 400 primary and middle schools in Sichuan, Jiangxi, and Sichuan, China. In total, 1,848 questionnaires were collected. After deleting those with missing data and those completing the survey in less than 3 min ($n = 589$), the survey resulted in 1,259 valid responses. There were 664 primary teachers and 595 middle schools, with a predominance of female teachers (primary schools: $n = 537$, 80.87%; middle schools: $n = 557$, 93.61%). There was no native English speaker among our participants. Their mother tongue is Chinese with English as their second language. Demographic details are presented in Table 1.

### Instruments

#### The Depression Anxiety Stress Scale-21

Teachers' PD was assessed using the Chinese version (*Chan et al., 2012*) of the 21-item Depression Anxiety Stress Scale (DASS-21; *Lovibond & Lovibond, 1995a*; *Lovibond &*

**Table 1 Key characteristics of the participants of the study (N = 1,259).**

| Demographic variable | Category | Value | Overall population statistics[a] |
|---|---|---|---|
| Gender; N (%) | | | $\chi^2 = 0.016$ ($p = 0.898$) |
| | Male | 165 (13.11%) | 155,786(12.98%) |
| | Female | 1,094 (86.89%) | 1,044,001(87.02%) |
| School stage; N (%) | | | $\chi^2 = 0.015$ ($p = 0.903$) |
| | Primary school | 595 (47.26%) | 569,084(47.43%) |
| | Junior high school | 664 (52.74%) | 630,703(52.57%) |
| Teaching experience; N (%) | | | Not available |
| | $\leq 5$ year | 398 (31.61%) | |
| | $>5$ years | 861 (68.39%) | |
| School type; N (%) | | | Not available |
| | Public school | 1,180 (94.73%) | |
| | Private school | 79 (6.27%) | |
| Headteacher; N (%) | | | Not available |
| | Yes | 618 (49.09%) | |
| | No | 641 (50.91%) | |

**Notes.**
[a] Data from the Ministry of Education of the People's Republic of China http://www.moe.gov.cn/jyb_sjzl/moe_560/2022/.

*Lovibond, 1995b*). This 4-point Likert scale (0 = not at all to 3 = most of the time) measures three dimensions: depression (*e.g.*, "I couldn't seem to experience any positive feeling at all"), anxiety (*e.g.*, "I was aware of dryness of my mouth"), and stress (*e.g.*, "I found it hard to wind down"). Each subscale contains seven items, with subscale scores calculated by doubling the sum of item responses (range = 0–21 per subscale). The total score ranges from 0 to 63, with higher scores indicating more severe PD. Following Lovibond and Lovibond's (*1995a*; *1995b*) established clinical thresholds, participants were classified as experiencing clinically significant levels of distress if they scored: $\geq 10$ on depression, $\geq 8$ on anxiety, or $\geq 15$ on stress subscales. The Chinese version of the DASS-21 has been well demonstrated by prior studies (*e.g.*, *Cao et al., 2023*; *Zhou et al., 2024*). In the present study, the results of confirmatory factor analysis (CFA) revealed an ideal fit for the three-factor structure (CFI = 0.929, TLI = 0.92, RMSEA = 0.047, SRMR = 0.036). The internal consistency of the scale was high (depression: $\omega = 0.962$; anxiety: $\omega = 0.956$; stress: $\omega = 0.951$).

### The Smartphone Application-Based Addiction Scale

The Chinese version (*Leung et al., 2020*) of the Smartphone Application-Based Addiction Scale (SABAS; *Csibi et al., 2018*) was used to measure teachers' level of smartphone addiction. The scale consists of six items rated on a 6-point Likert scale (1 = strongly disagree to 6 = strongly agree), with total scores ranging from 6 to 30. Example items include: "You feel an urge to use social media more and more" and "You use social media in order to forget about personal problems". Higher total scores reflect greater severity of smartphone addiction. The scale performed the sound psychometric properties in Chinese samples (*Chen et al., 2020a*; *Chen et al., 2020b*; *Leung et al., 2020*). The expected unidimensional structure of SABAS gained an acceptable model fit in our present study

(CFI = 0.998, TLI = 0.995, RMSEA = 0.074, SRMR = 0.028). Besides, the scale showed a good internal consistency ($\omega = 0.895$).

### The Bergen Social Media Addiction Scale

The Chinese version (*Leung et al., 2020*) of the Bergen Social Media Addiction Scale (BSMAS; *Andreassen et al., 2016*) was adopted to measure teachers' problematic social media use. The measure comprises six items rated on a 5-point Likert scale (1 = very rarely to 5 = very often), yielding total scores ranging from 6 to 36. Representative items include: "You spend a lot of time thinking about social media or planning how to use it" and "You feel an urge to use social media more and more". Higher total scores reflect more severe problematic social media use. The Chinese version of BSMAS has sound psychometric properties (*Chen et al., 2020a*; *Chen et al., 2020b*; *Leung et al., 2020*). The expected unidimensional structure of BSMAS gained an acceptable model fit in our present study (CFI = 0.981, TLI = 0.968, RMSEA = 0.066, SRMR = 0.027). Mcdonald's $\omega$ was 0.919, suggesting a good internal consistency.

### The internet Gaming Disorder Scale–short-form

The Chinese version (*Leung et al., 2020*) of the Internet Gaming Disorder Scale–Short-Form (IGDS9-SF; *Pontes & Griffiths, 2015*) evaluates teachers' level of internet gaming disorder. The instrument utilizes a 5-point Likert scale (1 = never to 5 = very often) comprising nine items assessing gaming disorder symptoms. Representative items include: "Do you feel more irritability, anxiety or even sadness when you try to either reduce or stop your gaming activity?" and "Do you systematically fail when trying to control or cease your gaming activity?". Total scores range from 9 to 45, with higher scores indicating greater severity of gaming disorder symptoms. The psychometric properties of the Chinese version of IGDS9-SF have been established (*Chen et al., 2020a*; *Chen et al., 2020b*; *Leung et al., 2020*). The expected unidimensional structure performed a good goodness of fit (CFI = 0.999, TLI = 0.999, RMSEA = 0.068, SRMR = 0.030). The high internal consistency ($\omega = 0.977$) supported the robust reliability of the scale.

## Data collection

Due to data availability, the study conducted a cross-sectional online survey targeting primary and middle school teachers in four provinces of China (Sichuan, Jiangxi, and Sichuan provinces), using Sojump, an online questionnaire platform. A non-probability sampling (convenience sampling) was used to recruit our participants. The data was collected between 25 May and 30 June 2020. Initially, we reached out to principals of primary and middle schools in the four provinces. Those who agreed to participate then distributed the survey link to their teachers. Prior to starting the survey, electronic consent was obtained from all participants. Only those who agreed to participate were given access to the main questionnaire. No data were collected or analyzed from individuals who declined to participate. This research project received approval from the Jiangxi Psychological Consultant Association's Institutional Review Board (IRB ref: JXSXL-2020-J013) and followed the ethical principles outlined in the Declaration of Helsinki.

**Table 2   Results of the latent profile analysis.**

| Number of profiles | AIC | BIC | Entropy | BLRT_$p$ |
|---|---|---|---|---|
| 2 | 16,387.38 | 16,438.76 | 0.97 | 0.01 |
| 3 | 14,937.05 | 15,008.98 | 0.98 | 0.01 |
| 4 | 14,945.08 | 15,037.56 | 0.77 | 1.00 |
| 5 | 14,953.04 | 15,066.08 | 0.68 | 0.01 |

Notes.

AIC, Akaike Information Criteria; BIC, Bayesian Information Criteria; BLRT_$p$, $p$- value of the Bootstrap Likelihood ratio test.

## Data analysis

We first excluded 589 invalid responses, including (1) all cases with missing data (which were found to have completion times under 3 min) ($n = 12$), and (2) additional cases with completion times below 3 min but no missing data ($n = 577$). This resulted in 1,259 valid responses. Then, we utilized LPA to identify participants' latent groups (or profiles). Based on the results of LPA, we conducted multivariate logistic regression to explore the demographic characteristics of different latent groups. After that, we employed one-way ANOVA to reveal the different levels of three types of PIU across various latent groups. The optimal number of profiles is determined by Akaike information criteria (AIC), Bayesian information criteria (BIC), entropy, and bootstrap likelihood ratio test (BLRT). Lower AIC and BIC better model fit. Higher entropy indicates better classification accuracy with a value above 0.80 indicating acceptable delineation of clusters (*Scotto Rosato & Baer, 2012*). A significant $p$-value (less than 0.05) in the BLRT suggests that a k class model improves the fit over the $k$-1 class model (*Spurk et al., 2020*). The number of profiles to be compared ranges from 2 to 6. We employed R (4.3.3) and SPSS (26.0; IBM Corp., Armonk, NY, USA) for statistical analyses.

# RESULTS

## The results of latent profile analysis

Table 2 presents the fit indices for different models with varying numbers of latent profiles. AIC and BIC decreased until the number of latent profiles was set at 3. The entropy value was 0. 98, exceeding the threshold of 0.8. Besides, the BLRT test of the 3-profile model was significant ($p < 0.001$) indicating that the 3-profile solution fitted the data better than the 2-profile model. Although the smallest group comprised only 44 teachers (3.49% of the sample), this group represented the subgroup who have the most severe PD in our sample and revealed important heterogeneity in the data that has significant practical implications (*Masyn, 2013*). Therefore, we retained this subgroup and selected the 3-profile model as the optimal solution.

Guided by Fig. 1C, we named the three profiles based on the scores for depression, anxiety, and stress. The scores for all three indicators in the first class (C1) exceeded 10, which, when doubled, far surpassed the diagnostic thresholds (10 for depression, 8 for anxiety, and 15 for stress). Consequently, this class was labeled as the "moderate-to-severe PD profile" ($n = 44$, 3.49%). Teachers in the second class (C2) had scores just above the

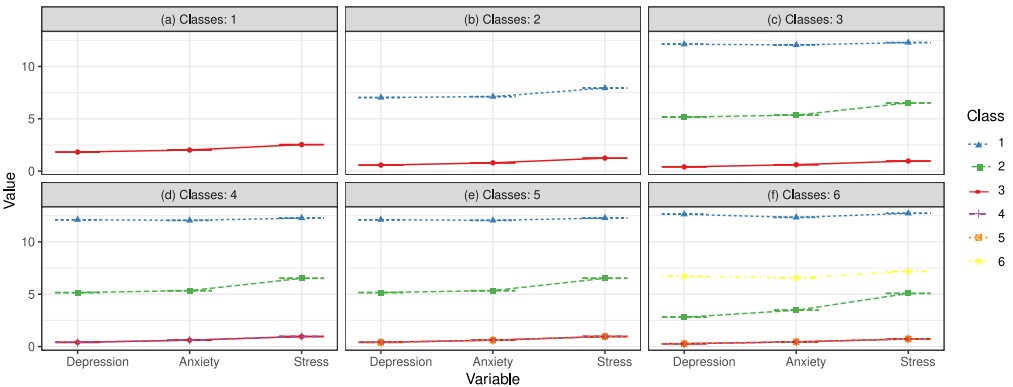

**Figure 1** The different models of LPA with various profiles.

**Table 3** The detection rate of the number of positive symptoms across profiles.

| Profile | | The detection rate of the number of positive symptoms | | | | Total |
|---|---|---|---|---|---|---|
| | | **0** | **1** | **2** | **3** | |
| C1 Moderate-to-Severe | N | 0 | 0 | 1 | 43 | 44 |
| | % | 0.00% | 0.00% | 2.27% | 97.73% | |
| C2 Mild | N | 19 | 85 | 129 | 29 | 262 |
| | % | 7.25% | 32.44% | 49.24% | 11.07% | |
| C3 Mentally Healthy | N | 935 | 18 | 0 | 0 | 953 |
| | % | 98.11% | 1.89% | 0.00% | 0.00% | |

thresholds, indicating mild PD. Accordingly, this class was labeled as the "mild PD profile" ($n = 262$, 20.81%). The third class (C3) scored very low on all three indicators, and we named this class the "mentally healthy profile" ($n = 953$, 75.69%).

Further examination of the outcomes of the 3-profile model demonstrated its strong discriminatory capability. As shown in Table 3, the detection rate for the number of positive symptoms in the "moderate-to-severe PD profile" (C1) was 100%. Over 97% of participants in this latent group exhibited high levels of positive symptoms of depression, anxiety, and stress simultaneously. The detection rate among teachers with the "mild PD profile" (C2) was 92.25%, with teachers in this group typically reporting one or two mild levels of PD. In contrast, the detection rate for the "mentally healthy profile" (C3) was only 1.89%. Most teachers in this group did not report any positive symptoms. In rare cases where positive symptoms were present, they were mild and singular. These findings indicate that the 3-profile model effectively identifies abnormal cases, further supporting the 3-profile solution.

## The results of multivariate logistic regression analysis

Based on the results of LPA, this study further explores the demographic characteristics of three different PD latent groups. We conducted a multivariate forward-stepwise logistic regression analysis with the "mentally healthy" group as the reference group. The dependent

**Table 4  The results of multivariate logistic regression analysis.**

|    | Variables | | B | SE | Wald | df | p | OR | 95%CI |
|----|-----------|--|-----|------|------|------|-------|------|--------------|
| C1 | Teaching experience | ≤5 years | −0.54 | 0.39 | 1.93 | 1.00 | 0.165 | 0.58 | (0.27, 1.25) |
|    |           | >5 years* | | | | | | | |
|    | Gender    | Male | 0.82 | 0.37 | 4.85 | 1.00 | 0.028 | 2.27 | (1.09, 4.71) |
|    |           | Female* | | | | | | | |
| C2 | Teaching experience | ≤5 years | −0.38 | 0.16 | 5.52 | 1.00 | 0.019 | 0.68 | (0.50, 0.94) |
|    |           | >5 years* | | | | | | | |
|    | Gender    | Male | 0.41 | 0.20 | 4.34 | 1.00 | 0.037 | 1.51 | (1.02, 2.21) |
|    |           | Female* | | | | | | | |

**Notes.**

OR, odds ratio; CI, confidence intervals.
Reference group: C3.
*reference category.

variable is the latent groups of teachers, and the independent variables include gender, teaching experience, school type (private school *vs* public school), school stages (primary school *vs* junior high school), and whether the teacher held the position of headteacher. The results indicated that only gender and teaching experience could enter the final model successfully and had a significant impact on the distribution of PD patterns among language teacher populations (see Supplementary Material).

The results (Table 4) indicated that novice teachers with less than or equal to five-year teaching experience are 0.68 times as likely as those experienced teachers with more than five-year teaching experiences to develop mild PD (OR = 0.68, 95% CI [0.50–0.94]). This means novice teachers are 32% less likely to experience mild PD than experienced teachers. As for the odds of developing moderate-to-severe PD, the two subgroups did not show a significant difference ($p = 0.165$). In terms of gender, male language teachers were 2.27 times more likely to experience moderate-to-severe (OR = 2.27, 95% CI [1.09–4.71]) and 1.51 times more likely to experience mild PD ($OR = 1.51$, 95% CI = [1.02–2.21]) than female language teachers. These findings highlight the potential influence of gender and teaching experience on PD among language teachers, but further research is needed to confirm these associations.

## The results of one-way ANOVA

One-way ANOVA was employed to find the association between different PD profiles and PIU. The results (Table 5) indicated significant differences across PD groups in levels of smartphone addiction ($F = 63.52$, $p < 0.001$, $\eta^2 = 0.09$), problematic social media use ($F = 53.98$, $p < 0.001$, $\eta^2 = 0.08$), and internet gaming disorder ($F = 115.55$, $p < 0.001$, $\eta^2 = 0.16$). *Post-hoc* comparisons revealed that teachers experiencing PD (C1 and C2) reported higher levels of smartphone addiction, problematic social media use, and internet gaming disorder compared to mentally healthy teachers (C3). Furthermore, teachers with moderate-to-severe PD (C1) scored significantly higher in internet gaming disorder than those with mild PD (C2).
**Table 5  The results of the ANOVA analysis.**

|  | Class | N | M | SD | F | $\eta^2$ | Post hoc (Tukey) |
|---|---|---|---|---|---|---|---|
| Smartphone Addiction | C1 | 44 | 21.09 | 7.16 | 63.52[*] | 0.09 | C1 >C3 C2 >C3 |
|  | C2 | 262 | 20.01 | 5.57 |  |  |  |
|  | C3 | 953 | 15.80 | 5.97 |  |  |  |
| Problematic Social Media Use | C1 | 44 | 17.93 | 5.71 | 53.98[*] | 0.08 | C1 >C3 C2 >C3 |
|  | C2 | 262 | 16.11 | 4.25 |  |  |  |
|  | C3 | 953 | 13.26 | 4.75 |  |  |  |
| Internet Gaming Disorder | C1 | 44 | 21.00 | 9.52 | 115.55[*] | 0.16 | C1 >C3 C2 >C3 C1 >C2 |
|  | C2 | 262 | 21.00 | 9.52 |  |  |  |
|  | C3 | 953 | 12.13 | 4.90 |  |  |  |

**Notes.**
[*]$p < 0.05$.

# DISCUSSION

This study examined the latent profiles of PD among language teachers and their associated demographic characteristics. Additionally, we investigated variations in PIU across these distinct psychological profiles.

Based on the scoring patterns for depression, anxiety, and stress, three latent profiles (or groups) were identified: the moderate-to-severe PD profile, the mild PD profile, and the mentally healthy profile. The frequency distribution of these latent groups resembled a pyramid: the more severe the PD, the fewer individuals fell into that group. Most of the participating language teachers were classified as mentally healthy, indicating most language teachers were mentally healthy. More than 20 percent of teachers in our samples have PD. Among these, teachers with mild PD exhibited a "gray" psychological state, requiring timely support and intervention to prevent further deterioration. As for those with moderate-to-severe PD, they are in a dangerous mental state, representing a high-risk population for psychological crisis events and warranting particular attention. Early and effective identification of such teachers is crucial for enabling relevant departments to implement measures that ensure the quality of teaching.

Further observation of the three latent profiles revealed that participants tended to report relatively comparable levels of depression, anxiety, and stress. This means that teachers experiencing one form of PD concurrently experience the other two. Such an observation aligns with previous studies that have consistently documented the concurrency of depression, anxiety, and stress (*Arvidsdotter et al., 2016*; *Vanden Bergh, Marchetti & Koster, 2021*). Traditionally, this phenomenon has been attributed to shared underlying causes, such as neuroticism or common environmental stressors (*Lovibond & Lovibond, 1995a*; *Lovibond & Lovibond, 1995b*). However, recent research offers a new perspective through the network approach. According to this approach, the concurrency does not stem from latent causes but rather from mutual reinforcement (or inhibition) of symptoms across the three dimensions of distress (*Vanden Bergh, Marchetti & Koster, 2021*). The symptom clusters of depression, anxiety, and stress are interconnected through

"bridges," which connect different clusters and facilitate the mutual reinforcement (or inhibition), providing a fundamental mechanism for understanding their concurrent occurrence (*Borsboom, 2017*; *Vanden Bergh, Marchetti & Koster, 2021*). Although *Mihić et al. (2024)* have identified symptoms such as agitation, restlessness, and an inability to relax as the "bridges" linking the three clusters in clinical samples, the concurrency of depression, anxiety, and stress among language teachers remains underexplored, deserving future further research.

The results of multivariate logistic regression analysis showed that gender and teaching experiences can significantly impact the distribution of different PD latent groups among language teachers, while school stages, school types, and whether the teacher held the position of head teacher cannot. Interestingly, we found experienced teachers (>5 years of teaching experience) have a higher likelihood of having PD, compared with less experienced teachers; Male language teachers are more likely to develop PD than their female counterparts. Different from the majority of the previous studies, this study indicated that an increasing teaching experience might not always indicate good mental health. This might partially be because online teaching during the Pandemic puts experienced teachers at a loss. Hence, facing the digital gap (*Ozoemena et al., 2021*), experienced teachers develop PD more easily than less experienced language teachers.

Opposed to our initial expectation, male language teachers have higher odds of having PD. This finding is inconsistent with the findings of most previous studies, which reported females are more vulnerable, due to the burden of caring responsibilities at home combined with their profession. However, given the profession is highly feminized, working in a predominantly female environment tends to cause ongoing barriers to job satisfaction and performance for males (*Cushman, 2005*), leading to male language teachers' PD. Besides, traditional masculine norms state that men should be stoic (*Vogel et al., 2011*) and inhibit male teachers from emotional expression and help-seeking (*Rochlen, McKelley & Pituch, 2006*), resulting in relatively severe PD. However, this inference needs to be taken with caution, because of the small number of our male participants, despite the gender ratio in our sample being nearly consistent with that in overall language teachers in China. Future studies are expected to focus on male language teachers to provide us with more detailed information about the phenomenon.

The results of one-way ANOVA revealed that teachers with PD scored significantly higher than mentally healthy teachers in the three examined issues about PIU: smartphone addiction, problematic social media use, and internet gaming disorder. Additionally, teachers with moderate-to-severe PD demonstrated significantly more severe internet gaming distress compared to those with mild PD. These findings align with previous studies (*Li et al., 2019*; *Pera, 2020*). On the one hand, teachers experiencing PD are more likely to seek a sense of satisfaction, control, and achievement through internet use to alleviate their negative experiences (*Morahan-Martin, 1999*; *Young, 2017*). As a result, they are at a higher risk of PIU compared to mentally healthy teachers (*He & Chen, 2024*). On the other hand, excessive internet use can heighten emotional sensitivity and exacerbate the deterioration of PD (*Gao & Chen, 2006*; *Senol Durak & Durak, 2013*), such as depression, anxiety, and stress, trapping teachers into the "PD–PIU" vicious cycle.

## IMPLICATIONS

This study yields significant theoretical and practical implications for understanding and addressing language teachers' PD and PIU. Theoretically, by systematically examining how varying degrees of PD correlate with PIU manifestations, we advance our understanding of the association between PD and PIU within language teaching contexts. The identification of distinct PD profiles with differential PIU susceptibility contributes novel empirical evidence to the growing body of research on occupational mental health in language education contexts. These findings establish a crucial foundation for future investigations into the mechanisms linking emotional labor and digital coping strategies among educators.

From a practical standpoint, our results offer concrete guidance for developing multi-level support systems. The delineation of different PD profiles suggests that interventions should move beyond generic approaches to incorporate profile-specific strategies. Educational institutions would benefit from implementing tiered support mechanisms, beginning with routine mental health screenings to identify at-risk individuals, followed by targeted professional development programs addressing digital literacy and stress management. At the organizational level, administrators should reconsider workload distribution models and institutionalize peer-support networks to mitigate professional distress factors contributing to PIU. The study further underscores the necessity of cross-sector collaboration in safeguarding educator well-being. Our findings corroborate *Jin & Yu's (2024)* call for coordinated societal engagement, emphasizing three critical action areas: policy reforms to establish mental health protection standards for educators, institutional initiatives to optimize working conditions and provide digital wellness training, and cultural shifts to foster more realistic public expectations of teaching professionals. By integrating these approaches, stakeholders can collectively construct a sustainable support ecosystem that not only addresses immediate PIU concerns but also enhances long-term professional satisfaction and teaching efficacy in language education.

## LIMITATIONS

However, this study has several limitations. First, due to practical constraints, the sample was drawn from specific regions in China, limiting the generalizability of the findings to broader populations. Second, as a cross-sectional study, it cannot establish causal relationships between PD and PIU. Future research can adopt longitudinal designs to examine the complex interplay between these constructs over time. Lastly, this study relied solely on quantitative methods, lacking qualitative data that could provide richer insights. Future studies could incorporate qualitative or mixed-method approaches to explore these issues in greater depth.

## CONCLUSION

The study identified three latent PD profiles: moderate-to-severe PD profile, mild PD profile, and mentally healthy profiles. Gender and teaching experience significantly predict the severity of language teachers' PD. Besides, our study also revealed that the more severe language teachers' PD, the higher their likelihood of experiencing PIU. These findings

underscore the importance of addressing language teachers' mental health needs, as timely intervention may help prevent the development of a detrimental "PD-PIU" cycle. Proactive measures targeting teachers' internet use behaviors may similarly disrupt this vicious cycle.

### Funding
This work was supported by Shanghai Open University Academic Team Cultivation Project "Research Team on the Continuous Development of TPACK for Open University Teachers" (2024TD004). The funders had no role in study design, data collection and analysis, decision to publish, or preparation of the manuscript.

### Grant Disclosures
The following grant information was disclosed by the authors:
Research Team on the Continuous Development of TPACK for Open University Teachers: 2024TD004.

### Competing Interests
The authors declare there are no competing interests.

### Author Contributions

- Zizheng Shen conceived and designed the experiments, performed the experiments, analyzed the data, prepared figures and/or tables, authored or reviewed drafts of the article, and approved the final draft.
- Honggang Liu conceived and designed the experiments, performed the experiments, analyzed the data, authored or reviewed drafts of the article, and approved the final draft.
- I-Hua Chen conceived and designed the experiments, analyzed the data, authored or reviewed drafts of the article, and approved the final draft.

### Human Ethics
The following information was supplied relating to ethical approvals (i.e., approving body and any reference numbers):

This research project received approval from the Jiangxi Psychological Consultant Association's Institutional Review Board (IRB ref: JXSXL-2020- J013).

### Data Availability
The raw data is available in the Supplementary File.

### Supplemental Information
Supplemental information for this article can be found online at http://dx.doi.org/10.7717/peerj.19707#supplemental-information.

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
