# Peer review of "Psychological distress and problematic internet use among language teachers: a latent profile analysis"

_PeerJ, doi:10.7717/peerj.19707_

## Round 0.1 · original submission · Major Revisions

Dear Dr. Shen and Dr. Liu,

Thank you for submitting your manuscript to PeerJ. Your paper was evaluated by two individuals with strong expertise in this area. The two reviewers have found your research question and study interesting and timely. At the same time, they have identified several concerns. Based on my independent reading, I agree that your paper has potential for an impactful contribution, but major revisions are necessary. Thus, I am inviting you the option of revising and resubmitting it for consideration.

The reviewers have provided exceptionally thorough and constructive comments. I am very grateful for their contributions. I will not repeat their points but I urge you to attend to all of their points, especially those related to the methodology, validity of the findings and your statistical analyses. You are welcome to submit a revision, if you think that you can address all concerns.

Thank you for giving us the opportunity to consider your work, and we wish you all the best with your article.

Yours sincerely,

Andree Hartanto, PhD
Academic Editor


Reviewer 1 ·

Basic reporting

-

Experimental design

-

Validity of the findings

-

Additional comments

The authors examined characteristics of language teachers using latent profile latent class analysis. The strength of the manuscript is that the topic is very original that offering mental health professionals important information about dealing with psychological well-being problems of students. I liked its originality very much. However, as an experienced editor and reviewer, I think that although the article is generally well-written, some comprehensive revisions must be made before publication of the article. I can summarize the main problems as follows, section by section, when possible, with suggestions.
Title
1. Page 5, Line 1-3: The title of the article must be revised to avoid ‘ in title. It is not recommended practice in the title. One revision may be that ''Psychological Distress and Problematic Internet Use Among Language Teachers: A Latent Profile Analysis’’ is straightforward and clear.
Abstract
2. Page 5, Line 24-25: In the following sentence ‘ ‘After that, ANOVA was conducted to examine the relationships between psychological distress and problematic internet…’’ ANOVA must change to one-way analysis of variance (ANOVA). ANOVA is a general term used for all types of ANOVA.
3. Page 5, Line 28-29: Following sentence ‘‘The results of multiple logistic regression indicated that male, experienced teachers were more likely to experience severe psychological distress.’’ must revise as ‘‘The results of multiple logistic regression indicated that male and experienced language teachers were more likely to experience severe psychological distress.’’
4. Page 5, Line 29-31: The following sentence needs revision for better clarity, and reported is not the correct way to give information about statistical findings. ‘‘The results of ANOVA reported that teachers experiencing psychological distress reported notably higher levels of problematic internet use, with those in the moderate-to-severe psychological distress group scoring particularly high in internet gaming disorder.’’ must revise as ‘ ‘ The results of one way ANOVA suggested that language teachers experiencing psychological distress reported notably higher levels of problematic internet use, with those in the moderate-to-severe psychological distress group scoring particularly high in internet gaming disorder.’’
5. Page 2, Line 31-33: The following sentence is unclear and needs revision. This reflected the close association between psychological distress and problematic internet use in language teachers.’’ One revision may be that ‘‘Results of this study indicate the close association between psychological distress and problematic internet use in language teachers.’’
6. Page 2, Line 38-39: Authors need to add the country to the Keywords.
7. Page 2, Line 23:
Introduction
8. Introduction, General: Because the dependent variable of study was psychological stress, the Introduction section must begin with a general introduction of the commonality of these symptoms among teachers and other school personnel, such as school counselors, and then must emphasize the language teachers. This introduction is much direct for this article. These additions can be made on Page 6, before Line 42. Some recent articles reference this topic, such as:
Agyapong, B., Obuobi-Donkor, G., Burback, L., & Wei, Y. (2022). Stress, burnout, anxiety, and depression among teachers: A scoping review. International journal of environmental research and public health, 19(17), 10706.
Ozamiz-Etxebarria, N., Idoiaga Mondragon, N., Bueno-Notivol, J., Pérez-Moreno, M., & Santabárbara, J. (2021). Prevalence of anxiety, depression, and stress among teachers during the COVID-19 pandemic: A rapid systematic review with meta-analysis. Brain sciences, 11(9), 1172.
9. Introduction, General: Each paragraph must consist of at least three to at most eight sentences as per APA 7 rules. Authors must correct the Introduction as well as the manuscript as per this rule. For example, Page 6 Line 42-66 is overly long.
10. Page 6, Line 58-59: remove two spaces between words in the following sentence: ‘Firstly, the study sought to explore the latent profiles of psychological distress in a cohort of language teachers in China and reveal the demographic characteristics of…’’
11. Page 6, Line 58-59: Authors need to provide long names of abbreviations in first citations, UNESCO
12. Page 8, Line 120: Authors must check the correctness of the following. This age range is the retirement age group in most countries. ‘‘Li et al. (2020) reported that male teachers aged 60 to 100 years had the highest prevalence of anxiety.’’
13. Page 8, Line 126-128: Authors must give more information about the advantages of using LCA to examine psychological stress among language teachers. Only one sentence is not enough.
14. Introduction, General: One main problem of the Introduction section is that authors a lot of repetitive content exist especially related to teacher psychological stress. Authors must carefully read the Introduction and avoid these.
15. Introduction, General: The second problem of the Introduction section is that authors must give information about the importance of this study in their cultural context. Specifically, why is it important and necessary to examine the association between psychological stress among problematic internet use in Chinese cultural contexts using LPA? This is an important reason for conducting this study. This information must be given in the introduction section, at least one or two paragraphs.
16. Introduction, General: The aim of the study is to be repeated twice. The beginning of the Introduction section must be removed or combined before the research questions are presented at the end of the introduction.
Method
17. Method, General: The Research design section is completely missing and must be added.
18. Page 9, Line 187-188: How many participants initially, this information must be added ‘ ‘The survey resulted in 1259 valid responses, after deleting those with missing data and those completing the survey in less than 3 minutes.’’ Moreover, deleting participants with missing values may lead to important bias in the analysis. Did the author examine missing data mechanisms?
19. Page 9, Line 186-194: Participants Section, General: Authors must add the age range of teachers with mean and standard deviations.
20. Page 9, Line 186-194: Participants Section, General: Authors must add used sampling design used.
21. Page 9, Line 186-194: Participants Section, General: Following must move the Data collection section. This research project received approval from the Jiangxi Psychological Consultation Association’s Institutional Review Board (IRB ref: JXSXL-2020-J013) and followed the ethical principles outlined in the Declaration of Helsinki.’’
22. Page 9, Line 186-194: Participants Section, General: All small n representing frequencies must be italic.
23. Page 9, Line 197-198: Following ‘ ‘The four-point Likert Chinese version of the Depression Anxiety Stress Scale-21 (DASS-21) (Chan et al., 2012) adopted’’ must revise as ‘ ‘The four-point Likert Chinese version of the Depression Anxiety Stress Scale-21 (DASS-21; Chan et al., 2012) used to….’’
24. Page 10, Line 201-203: The citation/citations needed for the following sentence. Participants are considered to experience psychological distress in a specific subdimension if their scores meet or exceed the recommended cutoff values set by the scale’s developers: 10 for depression, 8 for anxiety, and 15 for stress.’’
Lovibond, S. H., & Lovibond, P. F. (1995). Manual for the Depression Anxiety Stress Scales (2nd ed.). Psychology Foundation of Australia.
Lovibond, P. F., & Lovibond, S. H. (1995). The structure of negative emotional states: Comparison of the Depression Anxiety Stress Scales (DASS) with the Beck Depression and Anxiety Inventories. Behaviour Research and Therapy, 33(3), 335–343. https://doi.org/10.1016/0005-7967(94)00075-U
25. Page 10, Line 201-203: Authors need to add a space before and after = along the manuscript. ω=0.956, ω = 0.956.
26. Instruments section, general, Page 10, Line 206-241: For all scales, including DASS-21, authors must add possible scores, the meaning of higher scores, and sample items.
27. Instruments section, general, Page 10, Line 206-241: For all scales, authors must cite the original developers of scales. It is not correct not to cite them. They own them, not the translated researchers.
28. Page 10, Line 213-214: The following sentence is logically incorrect and must be revised. ‘The expected unidimensional structure of CFA gained an acceptable model fit in our present study. ’ as 'The expected unidimensional structure of SABAS gained an acceptable model fit in our present study’’
29. Page 10, Line 213-214: Following sentence must revise ‘McDonald’s ω is 0.919…’ as ‘McDonald’s ω was 0.919…’’
30. Instruments section, general, Page 10, Line 206-241: All reliability estimates must be reported in the past tense. For example, Page 10 Line 230-231 support must be supported.
31. Method section, general: Data analysis and collection section must be separate. Data collection, then Data analysis. Moreover, authors must give more information about the ethical aspects of the study.
32. Page 10, Line 236: Following sentence must report on Participants section ‘ ‘A non-probability sampling (convenience sampling) was used to recruit our participants.’’
33. Page 10, Line 236: How many participants were removed from data cleaning in the assumptions of statistical analyses check?, authors must give information about it. After data collection and cleaning,’’
34. Page 10, Line 263: Following sentence ‘R(4.3.3) and SPSS (26.0) were employed for our data analysis’ revise as ‘We employed R (4.3.3) and SPSS (26.0) for statistical analyses.’’
Results
35. Page 10- Page 12, Results section general: All small n representing subgroups must be italic.
36. Page 10- Page 12, Results section general: All p-values must be reported with three decimals in the text and tables.
37. Page 10- Page 12, Results section general: All 95% CI: must report as 95% CI =
38. Page 10- Page 12, Results section general: along the results section and the manuscript, all findings must be reported with two or three decimals consistently.
39. Page 10-Page 12, Results section general: All p values like this p = 0.000 must be corrected as p < 0.00
40. Discussion and others
41. Page 13, Page 15, Discussion general: No need subtitle in the discussion section and must be removed.
42. Page 13, Page 15, Discussion general: Authors must construct separate sections for Limitations, Practical implications, and Conclusion sections, and must move all related information to the related sections.
43. Page 13, Page 15, Discussion general: Authors must add a general introduction sentence to the Discussion before findings their results. This study investigated….
44. References, general: The Issue number was missing in most references and must be added.
45. In the tables, M and SD must be italic Mean must be M (italic).

·

Basic reporting

While there is a good discussion of why teachers in general are under psychological distress, the cause of language teachers' psychological stress and its relation to technology is more cursorily discussed. Further development of this section would make a positive contribution to the overall research. However, the psychological distress of language teachers is mentioned in line 83, and then the psychological distress of all teachers is discussed again. Addressing the psychological distress of all teachers first and then addressing psychological distress specific to language teachers would increase the readability of the study.

Why language teachers were included in this study and why language teachers' psychological stress and problematic internet use are important “in the context of language teaching” should be discussed in more detail.

Experimental design

In line 195, in the explanation of the data collection tools, giving 2-3 items as examples from the items in the data collection tools will facilitate the understanding.

Validity of the findings

-

---

## Round 0.2 · accepted · Accept

Dear Dr. Shen and Dr. Liu,,

I am pleased to advise that the above paper has now been accepted for publication in PeerJ. Thank you for giving the Journal the opportunity to publish your work. We are impressed with your paper and believe that it will contribute well to the literature. Well done!

Best Regards,
Andree

Reviewer 1 ·

Basic reporting

See my comments.

Experimental design

See my comments.

Validity of the findings

See my comments.

Additional comments

Thanks for opportunity review revised manuscript entitled ‘‘Psychological Distress and Problematic Internet Use Among Language Teachers: A Latent Profile Analysis’’. I would like the thanks to authors. They make a good job for improving quality of their manuscript. Authors revised the manuscript as I requested with a good will. In this form, Introduction reflects well the previous studies and study aim, Method section and Result section is correct, and Discussion section adequately synthesis to previous study findings and current study results. Overall, I have no further comment regarding to manuscript. I congratulate to authors and wish them success on their future endeavors.

·

Basic reporting

I believe that the revision requests to the article have been carried out completely and correctly by the authors. All processes were handled effectively by the authors and the necessary improvements were completed on the manuscript. For this reason, I would like to thank the authors of the article. I believe that the article can be published as it

Experimental design

I believe that the revision requests to the article have been carried out completely and correctly by the authors. All processes were handled effectively by the authors and the necessary improvements were completed on the manuscript. For this reason, I would like to thank the authors of the article. I believe that the article can be published as it is.

Validity of the findings

I believe that the revision requests to the article have been carried out completely and correctly by the authors. All processes were handled effectively by the authors and the necessary improvements were completed on the manuscript. For this reason, I would like to thank the authors of the article. I believe that the article can be published as it is.

Additional comments

I believe that the revision requests to the article have been carried out completely and correctly by the authors. All processes were handled effectively by the authors and the necessary improvements were completed on the manuscript. For this reason, I would like to thank the authors of the article. I believe that the article can be published as it is.